# The psychosocial dimensions of seborrheic dermatitis: A cross-sectional study on anxiety, personality, and quality of life

**Gamze Taş Aygar** ⓘ *, **Hanife Karataş, Elif Kaya, Nur Şeyma Borazan, Selda Pelin Kartal**

Department of Dermatology, Etlik City Hospital, Ankara, Turkey

* gamze_0890@hotmail.com

## Abstract

### Background

Seborrheic dermatitis (SD) is a chronic inflammatory skin disorder that predominantly affects sebaceous gland-rich areas, such as the scalp, face, and chest. It is clinically characterized by erythematous plaques and scaling. Although its pathogenesis is multifactorial, involving genetic predisposition, microbial colonization, and environmental triggers, the psychosocial dimensions of SD remain insufficiently explored. Given the chronic and visible nature of the disease, psychological factors like anxiety and personality traits may shape patients' perception of disease burden and their quality of life (QoL).

### Objectives

This study aimed to assess the relationship between anxiety levels, personality traits, and the clinical severity and quality of life in patients with seborrheic dermatitis. By evaluating these dimensions together, the study seeks to better understand the psychosocial impact of SD.

### Materials and methods

A cross-sectional study was conducted with 210 adult South Dakota patients. Disease severity was assessed using the Seborrheic Dermatitis Area and Severity Index (SDASI), while QoL was measured using the Dermatology Life Quality Index (DLQI). Anxiety was evaluated using the Beck Anxiety Inventory (BAI), and personality traits via the Bortner Personality Scale. Correlation analyses and regression models were employed.

### Results

SDASI demonstrated a strong positive correlation with Physician Global Assessment (Rho = 0.815, $p < 0.001$), confirming its validity as a clinical severity tool. DLQI exhibited moderate positive correlations with both BAI and anxiety severity (Rho = 0.465 and Rho = 0.365, $p < 0.001$), indicating that anxiety substantially contributes to

**Data availability statement:** All relevant data are within the manuscript and its Supporting Information files.

**Funding:** The author(s) received no specific funding for this work.

**Competing interests:** The authors have declared that no competing interests exist.

patient-perceived disease burden. In contrast, SDASI showed no significant associations with anxiety levels or personality traits (all $p > 0.05$), highlighting a dissociation between physician-rated severity and patient-reported quality of life. Weak correlations were observed between DLQI and female gender (Rho = 0.159, $p = 0.011$) as well as seasonality (Rho = 0.145, $p = 0.018$).

## Conclusion

Seborrheic dermatitis imposes a significant psychosocial burden that is more strongly linked to anxiety than to objective clinical severity. Our findings underscore the importance of integrating routine psychological screening—such as BAI assessment—into SD management to identify patients at risk of diminished quality of life. This study contributes novel evidence by concurrently evaluating clinical, psychological, and personality dimensions, reinforcing the need for a multidimensional, patient-centered approach to chronic inflammatory skin disorders.

## Introduction

Seborrheic dermatitis (SD) is a chronic, relapsing skin disorder affecting sebaceous-rich areas such as the scalp, face, and chest. It affects approximately 1–3% of immunocompetent individuals and is associated with various risk factors including genetic predisposition, seasonal changes, and the proliferation of *Malassezia* species. Patients frequently report flares triggered by environmental and psychological factors, especially stress [1,2].

The pathogenesis of SD involves epidermal barrier dysfunction and sebaceous gland activity, which facilitate *Malassezia* overgrowth and skin inflammation. These mechanisms, in combination with psychological stressors, can contribute to disease onset and exacerbation [1,3,4].

Recent research has highlighted the role of stress-related neuroendocrine responses in skin disorders. In SD, stress may alter sebaceous gland activity and immune responses, aggravating clinical symptoms [5–8]. However, the psychological burden of SD, including anxiety and the influence of personality traits, remains under-explored despite the condition's visibility and chronic course.

Personality structure, particularly the distinction between Type A and Type B traits, may influence how individuals perceive and cope with chronic illnesses. Type A individuals, who are more competitive and stress-prone, might be more vulnerable to anxiety, whereas Type B individuals tend to be more relaxed and resilient [9,10].

This study aims to investigate the relationship between SD severity, anxiety, personality traits, and quality of life. By examining these psychosocial dimensions together—using the Seborrheic Dermatitis Area and Severity Index (SDASI), the Beck Anxiety Inventory (BAI), the Bortner Personality Scale, and the Dermatology Life Quality Index (DLQI)—the study provides a more integrated understanding of SD's impact. Unlike previous research, it uniquely incorporates personality assessment via the Bortner Scale to better inform holistic, patient-centered management approaches.

## Materials and methods

This cross-sectional study comprised a total of 210 adult patients aged between 18 and 65 years who presented to the dermatology outpatient clinics of Ankara Etlik City Hospital between April 2024 and October 2024. Patients were included if they were 18 years or older and had a confirmed diagnosis of seborrheic dermatitis affecting the scalp, face, auricular and postauricular regions, chest, or back. Exclusion criteria included patients under the age of 18, pregnant women, and individuals with a history of psychiatric disorders. The study received ethical approval from the Ethics Committee for Scientific Research Evaluation at Ankara Etlik City Hospital (Reference number: AESH-BADEK-2024–218). All participants were informed about the study purpose and procedures, and provided written informed consent prior to participation. The study did not include minors. All procedures were conducted in accordance with the Declaration of Helsinki.

### Assessment tools

Seborrheic Dermatitis Area and Severity Index (SDASI):
    SDASI was used to objectively evaluate the severity and extent of seborrheic dermatitis. The tool assesses disease severity based on erythema, scaling, and infiltration, scored on a scale of 0 (none) to 4 (very severe). Additionally, the percentage of the affected area was scored from 1 to 6, based on the extent of skin involvement. The total SDASI score is calculated by summing the severity and area scores, with scores categorized as mild (0–4.2), moderate (4.3–8.4), and severe (8.5–12.6).

### Beck ANXIETY INVENTORY (BAI)

Anxiety levels were assessed using the Turkish version of the BAI, a validated self-report questionnaire [11]. The inventory consists of 21 items measuring physical and emotional manifestations of anxiety. Scores were categorized as mild (8–15), moderate (16–25), and severe (26–63). This tool provided valuable insights into the intensity and nature of anxiety symptoms in patients.

### Bortner personality scale

Personality traits were evaluated using the Turkish version of the Bortner Personality Scale, which assesses Type A and Type B personality patterns [12]. The scale uses bipolar statements to measure behavioral tendencies related to stress, urgency, and competitiveness. A total score below 100 indicated a Type B personality, while scores above 100 reflected a Type A personality. The total score for each participant ranged from 21 to 168.

### Statistical methods

All recorded data were analyzed using the Statistical Package for Social Sciences, version 27.0 (SPSS Inc., Armonk, NY, USA). The normality of numerical data distributions was assessed using the Kolmogorov–Smirnov and Shapiro–Wilk tests. Continuous variables with normal distribution were presented as mean ± standard deviation, while non-normally distributed variables were presented as median (interquartile range). Categorical variables were expressed as frequencies and percentages. The Mann–Whitney U test was used to compare two independent groups when the data did not follow a normal distribution. Pearson chi-square tests were employed to compare categorical variables (e.g., gender vs. SD severity). Due to the non-normal distribution of SDASI, DLQI, and other relevant psychometric scores (BAI, Bortner), Spearman's rank correlation coefficient (Spearman's rho) was used to assess the strength and direction of associations among these variables. Correlation strength was interpreted as follows: 0.00–0.19 = very weak; 0.20–0.39 = weak; 0.40–0.59 = moderate; 0.60–0.79 = strong; 0.80–1.00 = very strong. All statistical tests were two-tailed, and significance was set at $p < 0.05$.

## Results

### 1. Demographics and lifestyle factors

The demographic characteristics of the study participants are presented in Table 1. The Type A and Type B personality groups did not differ significantly in terms of age, gender, marital status, education level, occupation, BMI, smoking habits, or alcohol use (all $p > 0.05$).

**Table 1. Demographic features of the patient groups n = 210).**

|  | Type A personality (n = 128) | Type B personality (n = 82) | P |
|---|---|---|---|
| Age (median: IQR) | 28.5 (14.0) | 27.5 (18.0) | 0.578 |
| Gender (n/%) |  |  | 0.451 |
| Female | 63 (49.2) | 36 (43.9) |  |
| Male | 65 (50.8) | 46 (56.1) |  |
| Marital status (n/%) |  |  | 0.339 |
| Single | 65 (50.8) | 47 (57.3) |  |
| Married | 61 (47.7) | 32 (39.0) |  |
| Widow | 2 (1.6) | 3 (3.7) |  |
| Education (n/%) |  |  | 0.461 |
| Primary School | 6 (4.7) | 7 (8.5) |  |
| Secondary School | 30 (23.4) | 16 (19.5) |  |
| Bachelor's Degree | 92 (71.9) | 59 (72.0) |  |
| Occupation (n/%) |  |  | 0.839 |
| Non-working | 4 (3.1) | 2 (2.4) |  |
| Officer | 39 (30.5) | 27 (32.9) |  |
| Student | 32 (25.0) | 26 (31.7) |  |
| Others | 21 (16.4) | 13 (15.9) |  |
| Housewife | 14 (10.9) | 6 (7.3) |  |
| Worker | 9 (7.0) | 5 (6.1) |  |
| Retired | 9 (7.0) | 3 (3.7) |  |
| BMI (median: IQR) | 24.4 (5.0) | 24.6 (5.1) | 0.955 |
| Weight status |  |  | NA |
| Normal weight | 67 (52.3) | 42 (51.2) |  |
| Underweight | 3 (2.3) | 3 (3.7) |  |
| Overweight | 40 (31.3) | 27 (32.9) |  |
| Grade 1 obesity | 13 (10.2) | 7 (8.5) |  |
| Grade 2 obesity | 3 (2.3) | 2 (2.4) |  |
| Grade 3 obesity | 2 (1.6) | 1 (1.2) |  |
| Smoking (n/%) |  |  | 0.161 |
| Non-smoker | 74 (57.8) | 58 (70.7) |  |
| Smoker | 46 (35.9) | 21 (25.6) |  |
| Gave-up smoking | 8 (6.3) | 3 (3.7) |  |
| Alcohol (n/%) |  |  | 0.421 |
| Non-user | 96 (75.0) | 67 (81.7) |  |
| User | 31 (24.2) | 15 (18.3) |  |
| Gave-up alcohol | 1 (0.8) | – |  |

IQR: Interquartile range.

## 2. Clinical characteristics

Table 2 summarizes the clinical features of seborrheic dermatitis in both personality groups. There were no statistically significant differences between Type A and Type B groups in terms of systemic comorbidities, symptom type, location of lesions, trigger factors, or seasonal pattern of the disease ($p > 0.05$).

As shown in Table 3, disease duration, continuous vs. episodic course, number of flare-ups per year, duration of flare-ups, SDASI score and severity categories, Physician Global Assessment scores, DLQI, BAI scores, and anxiety status distribution also did not differ significantly between personality groups ($p > 0.05$). Notably, 30.9% of our patients exhibited moderate-to-severe anxiety, highlighting a substantial psychological burden in the overall cohort.

## 3. Correlations between variables

A strong positive correlation was observed between SDASI and Physician Global Assessment (Rho = 0.815, $p < 0.001$), supporting the validity of SDASI in assessing disease severity. SDASI was weakly positively correlated with scalp involvement (Rho = 0.149, $p = 0.031$), duration of disease (Rho = 0.160, $p = 0.010$), continuous course of SD (Rho = 0.238, $p < 0.001$), and number of flare-ups per year (Rho = 0.235, $p < 0.01$).

DLQI was weakly positively correlated with Physician Global Assessment (Rho = 0.157, $p = 0.011$), female gender (Rho = 0.159, $p = 0.011$), and seasonal pattern (Rho = 0.145, $p = 0.018$).

## 4. Associations with SDASI and DLQI severity

There were no significant differences in anxiety level, BAI score, personality type, or Bortner score according to SDASI severity categories ($p > 0.05$). Similarly, DLQI scores did not differ significantly based on number or location of involved skin areas.

## 5. Anxiety and personality traits

Moderate positive correlations were found between DLQI and both BAI (Rho = 0.465, $p < 0.001$) and degree of anxiety (Rho = 0.365, $p < 0.001$), indicating that greater anxiety was associated with worse quality of life. However, SDASI was not significantly correlated with BAI, anxiety severity, or personality traits (all $p > 0.05$). Personality type also did not show a significant correlation with either SDASI or DLQI.

## Discussion

This study investigated the relationship between psychological and personality factors, clinical severity, and perceived burden in SD, emphasizing the divergence between clinician-rated severity and patient-reported outcomes. Our findings support the multifactorial nature of SD, showing that objective clinical severity (SDASI) was not significantly associated with anxiety or personality traits, whereas perceived burden (DLQI) was moderately correlated with anxiety levels. This dissociation highlights the relevance of incorporating psychosocial parameters in the assessment of SD, as emotional distress may not be reflected in clinical evaluation alone.

The absence of significant associations between SDASI and psychological variables suggests that clinical severity in SD is likely driven by biological mechanisms such as sebaceous gland hyperactivity, immune dysregulation, and epidermal barrier impairment [1,7,13]. In contrast, the observed impact of anxiety on quality of life—independent of visible disease severity—is consistent with prior research indicating that patients' psychological state substantially influences their perception of disease burden [14–16].

Our findings are also consistent with previous literature emphasizing the psychological impact of SD. Baş et al. (2015) reported elevated anxiety and depression levels and impaired emotional and social quality of life in SD patients, though they did not assess associations with disease severity [17]. Cömert et al., including a control group, found significantly

**Table 2. Clinical features of the patient groups.**

| | Type A personality (n = 128) | Type B personality (n = 82) | P |
|---|---|---|---|
| Systemic diseases (n/%) | | | NA |
| None | 99 (77.3) | 67 (81.7) | |
| Hypertension | 2 (1.6) | 4 (4.9) | |
| Hypo/hyperthyroidism | 6 (4.7) | 4 (4.9) | |
| Diabetes Mellitus | 4 (3.1) | – | |
| Inflammatory diseases | 4 (3.1) | 2 (2.4) | |
| Others | 13 (10.2) | 5 (6.1) | |
| Erythema (n/%) | | | 0.564 |
| Absent | 28 (21.9) | 13 (15.9) | |
| Mild | 58 (45.3) | 36 (43.9) | |
| Moderate | 35 (27.3) | 29 (35.4) | |
| Severe | 7 (5.5) | 4 (4.9) | |
| Very severe | 0 (0.0) | 0 (0.0) | |
| Squamation (n/%) | | | 0.344 |
| Absent | 1 (0.8) | 3 (3.7) | |
| Mild | 54 (42.2) | 31 (37.8) | |
| Moderate | 57 (44.5) | 34 (41.5) | |
| Severe | 16 (12.5) | 13 (15.9) | |
| Very severe | 0 (0.0) | 1 (1.2) | |
| Infiltration (n/%) | | | 0.207 |
| Absent | 65 (50.8) | 31 (37.8) | |
| Mild | 39 (30.5) | 36 (43.9) | |
| Moderate | 16 (12.5) | 11 (13.4) | |
| Severe | 8 (6.3) | 4 (4.9) | |
| Very severe | 0 (0.0) | 0 (0.0) | |
| Symptoms | | | 0.378 |
| Asymptomatic | 11 (8.6) | 8 (9.8) | |
| Itching | 97 (75.8) | 55 (67.1) | |
| Burning and stabbing pain | 2 (1.6) | 4 (4.9) | |
| Itching with burning and stabbing Pain | 18 (14.1) | 15 (18.3) | |
| Location (n/%) | | | NA |
| Scalp | 109 (85.2) | 79 (96.3) | |
| Face | 13 (10.2) | 11 (13.4) | |
| Chest | 0 (0.0) | 4 (4.9) | |
| Ear | 59 (46.1) | 36 (43.9) | |
| Back | 44 (34.4) | 25 (30.5) | |
| Triggers (n/%) | | | NA |
| None | 4 (3.1) | 7 (8.5) | |
| Menstruation | 14 (10.9) | 4 (4.9) | |
| Food | 13 (10.2) | 5 (6.1) | |
| Stress | 116 (90.6) | 65 (79.3) | |
| Humidity | 13 (10.2) | 7 (8.5) | |
| Cosmetics | 16 (12.5) | 16 (19.5) | |
| Sun | 15 (11.7) | 7 (8.5) | |
| Infection | 4 (3.1) | 4 (4.9) | |

*(Continued)*

**Table 2.** (Continued)

| | Type A personality (n = 128) | Type B personality (n = 82) | P |
|---|---|---|---|
| Seasonal character | | | 0.279 |
| None | 50 (39.1) | 26 (31.7) | |
| Spring | 13 (10.2) | 15 (18.3) | |
| Summer | 40 (31.3) | 26 (31.7) | |
| Autumn | 17 (13.3) | 12 (14.6) | |
| Winter | 20 (15.6) | 17 (20.7) | |

**Table 3. Clinical features of the patient groups on seborrheic dermatitis.**

| | Type A personality (n = 128) | Type B personality (n = 82) | P |
|---|---|---|---|
| Duration of disease (years; median: IQR) | 5.0 (8.0) | 4.0 (4.3) | 0.096 |
| Continuous/Episodic | | | 0.5 |
| Continuous | 48 (37.5) | 27 (32.9) | |
| With episodes | 80 (62.5) | 55 (67.1) | |
| Number of episodes per year (median: IQR) | 5.0 (6.0) | 5.0 (6.0) | 0.944 |
| Dıration of attacks (days; median: IQR) | 14.0 (23.0) | 10.0 (23.0) | 0.12 |
| SDASI (median: IQR) | 5.2(3.0) | 5.4 (4.0) | 0.735 |
| SDASI groups (n/%) | | | 0.94 |
| Mild | 48 (37.5) | 32 (39.0) | |
| Moderate | 64 (50.0) | 39 (47.6) | |
| Severe | 16 (12.5) | 11 (13.4) | |
| Physician global assessment | | | 0.646 |
| 1 | 35 (27.3) | 28 (34.1) | |
| 2 | 66 (51.6) | 34 (41.5) | |
| 3 | 17 (13.3) | 14 (17.1) | |
| 4 | 9 (7.0) | 5 (6.1) | |
| 5 | 1 (0.8) | 1 (1.2) | |
| DLQI (median:IQR) | 9.0 (8.0) | 9.0 (8.0) | 0.518 |
| BAI (median:IQR) | 9.0 (17.0) | 9.0 (14.0) | 0.281 |
| Anxiety status (n/%) | | | 0.204 |
| Mild | 85 (66.4) | 60 (73.2) | |
| Moderate | 23 (18.0) | 16 (19.5) | |
| Severe | 20 (15.6) | 6 (7.3) | |

IQR: Interquartile range, SDASI: Seborrheic Dermatitis Area and *Severity* Index*, DLQI:* Dermatology Life Quality Index, BAI: Beck anxiety index.

higher anxiety prevalence in SD patients (32.5%) compared to healthy controls (12.6%), but no correlation with disease severity. Our results reflect this pattern, revealing no relationship between anxiety and SDASI, but a moderate correlation with DLQI, reinforcing the distinction between clinical observation and subjective burden.

Further supporting these observations, a recent meta-analysis reported clinically significant anxiety in 19% of SD patients—comparable to atopic dermatitis (21%) and acne vulgaris (30%) [18]. In our cohort, 30.9% of participants exhibited moderate-to-severe anxiety (BAI ≥ 16), exceeding the meta-analytic prevalence and closely paralleling the rate reported by Cömert et al [14]. Although our study did not include a control group, these comparisons underscore the

substantial psychological burden associated with SD. By simultaneously evaluating clinical severity, psychological distress, and personality traits in a relatively large sample, our study contributes to a more comprehensive understanding of SD and reinforces the need for integrated psychosocial assessment in dermatological care.

In addition to anxiety, we examined personality traits, focusing on Type A vs. Type B behavioral patterns. No significant associations were found between personality and disease severity or quality of life. However, prior studies have indicated that certain Type A traits—especially hostility and urgency—may be linked to systemic inflammation and stress-related pathophysiology [19–21]. In contrast, non-hostile traits such as conscientiousness may offer protective effects [22,23]. Although no direct relationship was observed in our sample, the influence of personality on stress perception and coping strategies warrants further study in SD.

Our results also provide insights into environmental and temporal disease patterns. Most patients reported episodic rather than continuous disease courses, with 63.9% citing seasonal triggers, particularly during summer. This aligns with literature suggesting more frequent flare-ups in warm, humid climates [24]. While studies in tropical settings have described chronic-recurrent patterns [24,25], our findings suggest that environmental variability in continental climates may influence intermittent disease activity. Stress was identified as the most frequent exacerbating factor, emphasizing its dual role as both a psychological and physiological trigger.

The significant correlation between Physician Global Assessment and SDASI supports SDASI's utility as a reliable measure of clinical severity. However, the weak correlation between SDASI and DLQI highlights the discrepancy between clinical evaluation and patient-perceived burden. Our findings support incorporating patient-reported outcomes such as the DLQI into routine dermatological assessment to ensure holistic and patient-centered care. Additionally, modest correlations of DLQI with female gender and seasonal sensitivity suggest that demographic and contextual factors may further modulate quality of life in SD.

This study has several limitations. First, its cross-sectional design prevents conclusions about causality between psychological factors and disease severity. Second, the sample was drawn from a single outpatient center, potentially limiting generalizability. Third, reliance on self-reported questionnaires may introduce response bias. While patients with diagnosed psychiatric disorders were excluded, unmeasured factors—such as socioeconomic status, lifestyle, and broader psychosocial context—were not assessed. Lastly, the absence of a control group limits direct comparison to the general population.

## Conclusion

This study elucidates the multifactorial nature of seborrheic dermatitis, revealing that the condition is shaped by biological, psychological, and environmental influences. While clinical severity (SDASI) showed no significant associations with anxiety or personality traits, patient-perceived burden (DLQI) was moderately correlated with anxiety severity (Rho = 0.465, p < 0.001), highlighting the dissociation between physician-rated disease and patient experience. These findings underscore the importance of incorporating systematic psychosocial screening into SD management. Routine use of anxiety screening tools (e.g., BAI) alongside conventional clinical assessments may help identify patients in need of psychosocial support, improving patient-centered care. By concurrently evaluating clinical, psychological, and personality dimensions in a single SD cohort, this study contributes novel insights to the dermatological literature and reinforces the value of a multidimensional, biopsychosocial approach in managing chronic inflammatory skin conditions.

## Supporting information

**S1 File. Data.**
(XLSX)

## Author contributions

**Conceptualization:** Hanife Karatas, Elif Kaya.

**Data curation:** Gamze Taş Aygar, Hanife Karatas, Elif Kaya, Nur Şeyma Borazan.

**Formal analysis:** Gamze Taş Aygar.

**Investigation:** Gamze Taş Aygar, Nur Şeyma Borazan.

**Methodology:** Gamze Taş Aygar, Nur Şeyma Borazan.

**Project administration:** Gamze Taş Aygar, Selda Pelin Kartal.

**Writing – original draft:** Gamze Taş Aygar.

**Writing – review & editing:** Selda Pelin Kartal.

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
