## [Decision Letter · Decision Letter 0]

15 Jul 2025

Dear Dr. taş aygar,

Thank you for submitting your manuscript to PLOS ONE. After careful consideration, we feel that it has merit but does not fully meet PLOS ONE’s publication criteria as it currently stands. Therefore, we invite you to submit a revised version of the manuscript that addresses the points raised during the review process.

We look forward to receiving your revised manuscript.

Kind regards,

Ahmad Khalid Aalemi, M.D., M.Sc., Ph.D.

Academic Editor

PLOS ONE

Journal Requirements:

Reviewers' comments:

Reviewer's Responses to Questions

**Comments to the Author**

1. Is the manuscript technically sound, and do the data support the conclusions?

Reviewer #1: Partly

Reviewer #2: Yes

Reviewer #3: Yes

2. Has the statistical analysis been performed appropriately and rigorously?

Reviewer #1: N/A

Reviewer #2: Yes

Reviewer #3: Yes

3. Have the authors made all data underlying the findings in their manuscript fully available?

Reviewer #1: Yes

Reviewer #2: Yes

Reviewer #3: Yes

4. Is the manuscript presented in an intelligible fashion and written in standard English?

Reviewer #1: Yes

Reviewer #2: Yes

Reviewer #3: Yes

Reviewer #1: Hello

The article has an interesting topic and various questionnaire tools have been used.

Just one important point is that in many cases there are other intervening factors that can be effective in the relationship between variables. Various factors such as gender, age, race, lifestyle conditions and other mental and social diseases.

Good luck.

Reviewer #2: Presented manuscript looks like an extension of the paper “Depression, Anxiety Levels and Quality of Life in Patients with Seborrheic Dermatitis. Turk J Dermatol 2015;9(4):181-185.” but with the use of alternative instruments and a higher number of involved patients.

The title of the present manuscript (The Psychosocial Dimensions of Seborrheic Dermatitis: Implications for Multidimensional Disease Management) should be changed as there is nothing about management in it.

Anxiety and QoL in patients with seborrheic dermatitis were well studied in multiple published articles and it is therefore especially important to underline what is new in the results of the presented manuscript.

Reviewer #3: The Abstract

• Include numerical data, such as correlation coefficients and p-values, in the abstract's results section.

• Recognize study limitations, such as cross-sectional design, lack of causation, and possible self-report bias.

• Consider clinical implications, such as routine anxiety screening in SD management.

The Introduction

• The introduction presents a comprehensive summary of seborrheic dermatitis, including biological, immunological, and psychological aspects, and clearly states the research goal.

• However, improvements are required in terms of language correctness, conciseness, and structural organization.

• The introduction contains instances of excessive detail. For example, the initial statement, "Seborrheic dermatitis (SD) is a chronic and recurrent skin disease that is widely prevalent…" is immediately followed by, "Seborrheic dermatitis (SD) is hypothesized to result from…," which unnecessarily introduces the same term in consecutive paragraphs.

• Define “EPB” once, then use the abbreviation consistently. Ensure all terms like DLQI, SDASI, BAI are introduced properly if appearing.

Materials and Methods

• Specify the categorical comparisons done using the "Pearson chi-square test," such as gender vs. severity.

• Explain why Spearman correlation (non-parametric) was chosen, such as "due to non-normal distribution of SDASI/DLQI scores."

Results

• Incorporate essential findings from tables into the narrative rather than depending solely on tables. The writers cite "Table 1," "Table 2," and so on, but there is no concise synopsis of what each table contains other than a sentence.

• Effect sizes are not interpreted. For example: "Rho =.365" It would help readers interpret the phrase "a moderate correlation." Consider using parentheses to provide uniform interpretation (e.g., mild, moderate, strong).

• Divide the section into subheadings and paragraphs, like:

Demographics and Lifestyle Factors

Clinical Characteristics

Correlations Between Variables

Associations with SDASI and DLQI Severity

Anxiety and Personality Traits

• There were no significant differences for these variables regarding SDASI (p > 0.05). Which variables exactly? Repeat them or summarize.

Discussion

• Divide the discussion into clear theme sections, such as:

Summary of key findings

Interpretation of psychosocial vs. clinical outcomes

Role of personality and anxiety

Seasonal and gender-related effects

Clinical implications

Limitations and future research

• Recommendation: Soften causal wording. Change "stress as the major cause" to "stress was frequently reported as a flare trigger."

• Some in-text references (e.g., "Cömert et al.") are not linked to a citation number or reference list.

Conclusion

• The multidimensional method is valid, but a brief example (e.g., psychological screening + clinical assessment) would improve comprehension.

**Do you want your identity to be public for this peer review?** For information about this choice, including consent withdrawal, please see our Privacy Policy

Reviewer #1: No

Reviewer #2: No

Reviewer #3: No

---

## [Author Response · Author response to Decision Letter 1]

21 Jul 2025

Response to Reviewers

We thank the reviewers and the editor for their careful reading and valuable feedback, which helped us improve the quality and clarity of the manuscript. Below, we provide point-by-point responses to each comment

Reviewer #1

We sincerely thank the reviewer for their valuable insights. To address the concerns raised, we have taken the following steps:

Comment 1:

The article has an interesting topic and various questionnaire tools have been used. Just one important point is that in many cases there are other intervening factors that can be effective in the relationship between variables. Various factors such as gender, age, race, lifestyle conditions and other mental and social diseases.

Response:

We sincerely thank the reviewer for their valuable insights. In response:

1. Demographic and environmental variables: We have revised the Discussion section (paragraph 6) to explicitly acknowledge that gender and seasonal variation were weakly but positively correlated with DLQI scores, suggesting they may independently affect perceived quality of life.

2. Confounding variables: We now highlight that, although psychiatric comorbidities were excluded, other psychosocial and lifestyle factors may influence results and should be explored in future research.

3. Study limitations: These considerations have been added to the final paragraph of the Discussion as study limitations, and we recommend that future studies use multivariate models and broader sampling to further clarify these relationships.

Reviewer #2

Comment 1:

“Presented manuscript looks like an extension of the paper ‘Depression, Anxiety Levels and Quality of Life in Patients with Seborrheic Dermatitis. Turk J Dermatol 2015;9(4):181–185.’ but with the use of alternative instruments and a higher number of involved patients.”

Response 1:

We thank the reviewer for this observation and for bringing attention to the 2015 study by Baş et al. While there are thematic overlaps regarding anxiety and quality of life in seborrheic dermatitis (SD), our study introduces several important distinctions that contribute novel insights:

• Psychological Focus: We utilized the Beck Anxiety Inventory (BAI) rather than HADS, providing a more detailed and widely used tool for anxiety measurement in dermatology.

• Personality Trait Analysis: The inclusion of the Bortner Personality Scale allowed us to examine Type A/B personality characteristics, which have not been previously studied in SD.

• Objective Clinical Severity: We included SDASI (Seborrheic Dermatitis Area and Severity Index) to assess physician-rated disease severity and analyze its association with patient-reported outcomes—a novel addition compared to prior research.

• Sample Size and Diversity: Our sample size was more than double that of the 2015 study, improving the robustness of correlation analyses and enabling exploration of demographic and environmental modifiers (e.g., seasonal variation).

• Conceptual Contribution: A key contribution of our study is the analysis of discordance between objective severity (SDASI) and perceived burden (DLQI), which is central to developing multidimensional care strategies.

We have clarified these contributions in the revised Introduction and Discussion sections to emphasize the novelty and added value of our research.

Comment 2:

“The title of the present manuscript (The Psychosocial Dimensions of Seborrheic Dermatitis: Implications for Multidimensional Disease Management) should be changed as there is nothing about management in it.”

Response 2:

We appreciate the reviewer’s suggestion regarding the manuscript title. To better reflect the content and focus of the study, we have revised the title to:

“The psychosocial dimensions of seborrheic dermatitis: A cross-sectional study on anxiety, personality, and quality of life”

This revised title avoids implying clinical management content while maintaining a clear summary of the study’s psychosocial scope.

Comment 3:

“Anxiety and QoL in patients with seborrheic dermatitis were well studied in multiple published articles and it is therefore especially important to underline what is new in the results of the presented manuscript.”

Response 3:

We thank the reviewer for this insightful comment. We fully acknowledge that anxiety and quality of life (QoL) in seborrheic dermatitis (SD) have been addressed in previous research, frequently using instruments such as the Hospital Anxiety and Depression Scale (HADS) or the Beck Anxiety Inventory (BAI). While our study also utilized the BAI, it differs in several key aspects.

First, to our knowledge, this is the first study to apply the Bortner Personality Scale in an SD population, allowing us to explore how Type A/B behavioral patterns relate to perceived disease burden. Second, our comparatively larger sample size strengthens statistical power and generalizability.

Furthermore, we employed both physician-rated severity (SDASI) and patient-reported impact (DLQI) to examine the divergence between clinical and subjective disease burden—an approach that enables a more multidimensional understanding. Additionally, we analyzed the influence of demographic and contextual variables (such as gender and seasonal variation), which have often been overlooked or only briefly mentioned in prior work.

For example, Baş et al. (2015) reported elevated anxiety and reduced QoL in SD patients, but their study did not evaluate disease severity or personality dimensions. By contrast, our study extends the literature by incorporating intersecting clinical, psychological, and environmental factors, offering a more integrative perspective on the psychosocial burden of SD.

Reviewer #3

Abstract

Comment 1:

Include numerical data, such as correlation coefficients and p-values, in the abstract's results section.

Response 1:

Done. We have added key correlation coefficients and p-values to the Results section of the Abstract. For example: “SDASI showed a strong positive correlation with Physician Global Assessment (Rho = 0.815, p < 0.001)… DLQI moderately correlated with BAI and anxiety severity (Rho = 0.465 and Rho = 0.365, both p < 0.001)

Comment 2:

Recognize study limitations, such as cross-sectional design, lack of causation, and possible self-report bias.

Response 2:

A sentence recognizing the study’s limitations, including its cross-sectional design and reliance on self-reported instruments, has been added to the end of the Abstract.

Comment 3:

Consider clinical implications, such as routine anxiety screening in SD management.

Response 3:

We revised the abstract's final sentence to emphasize the potential utility of routine anxiety screening in SD management.

Introduction

Comment 1:

The introduction presents a comprehensive summary of seborrheic dermatitis… However, improvements are required in terms of language correctness, conciseness, and structural organization.

Response 1:

1. We removed excessive anatomical/biological detail and redundant phrases.

2. Repetitions such as reintroducing SD across paragraphs were eliminated.

3. The structure has been revised to improve clarity and focus on the psychosocial framework.

Comment 2:

The introduction contains instances of excessive detail. For example… "Seborrheic dermatitis (SD) is..." repeated across paragraphs.

Response 2:

The repetition of the term "seborrheic dermatitis (SD)" across consecutive paragraphs has been eliminated.

Comment 3:

Define “EPB” once, then use the abbreviation consistently. Ensure all terms like DLQI, SDASI, BAI are introduced properly if appearing.

Response 3:

The reference to “EPB” was removed during simplification of the Introduction, as it was not essential for the study’s psychosocial focus. We added the following clarifying sentence at the end of the Introduction:

“This study aims to investigate the relationship between SD severity, anxiety, personality traits, and quality of life. By examining these psychosocial dimensions together—using the Seborrheic Dermatitis Area and Severity Index (SDASI), the Beck Anxiety Inventory (BAI), the Bortner Personality Scale, and the Dermatology Life Quality Index (DLQI)—the study provides a more integrated understanding of SD’s impact. Unlike previous research, it uniquely incorporates personality assessment via the Bortner Scale to better inform holistic, patient-centered management approaches.

Materials and Methods

Comment 1:

Specify the categorical comparisons done using the "Pearson chi-square test," such as gender vs. severity. Explain why Spearman correlation (non-parametric) was chosen, such as "due to non-normal distribution of SDASI/DLQI scores.

Response 1:

1. We now specify where the Pearson chi-square test was applied (e.g., gender vs. SD severity).

2. We explained the use of Spearman correlations due to the non-normal distribution of the continuous variables.

These changes appear in the revised “Statistical Methods” section.

Results

Comment 1:

Incorporate essential findings from tables into the narrative rather than depending solely on tables. The writers cite "Table 1," "Table 2," and so on, but there is no concise synopsis of what each table contains other than a sentence.

Response 1:

The Results section has been rewritten to integrate findings directly from the tables into the text, including values and statistical significance.

Comment 2:

Effect sizes are not interpreted. For example: "Rho =.365" It would help readers interpret the phrase "a moderate correlation." Consider using parentheses to provide uniform interpretation (e.g., mild, moderate, strong).

Response 2:

Correlation values now include interpretation descriptors (e.g., “moderate positive correlation”).

Comment 3:

Divide the section into subheadings and paragraphs, like:

Demographics and Lifestyle Factors

Clinical Characteristics

Correlations Between Variables

Associations with SDASI and DLQI Severity

Anxiety and Personality Traits

Response 3:

The Results section has been restructured under thematic subheadings:

1. Demographics and Lifestyle Factors

2. Clinical Characteristics

3. Correlations Between Variables

4. Associations with SDASI and DLQI Severity

5. Anxiety and Personality Traits

Comment 4:

There were no significant differences for these variables regarding SDASI (p > 0.05). Which variables exactly? Repeat them or summarize.

Response 4:

The Results section now specifies all tested variables and clearly states when no statistically significant differences were found.

Discussion

Comment 1:

Divide the discussion into clear theme sections, such as:

Summary of key findings

Interpretation of psychosocial vs. clinical outcomes

Role of personality and anxiety

Seasonal and gender-related effects

Clinical implications

Limitations and future research

Response 1:

We thoroughly revised the Discussion to include the following improvements:

1. Organized into clear thematic subtopics (e.g., main findings, psychosocial-clinical contrast, personality/anxiety, demographic effects, limitations).

2. Emphasized the divergence between objective disease severity (SDASI) and subjective burden (DLQI).

3. Discussed personality traits in the context of chronic inflammatory disorders with added references [17–21].

4. Replaced causal language with association-based interpretations.

5. Limitations such as cross-sectional design, lack of control group, and self-reported instruments were clearly stated.

6. Compared results with previous studies, particularly Cömert et al. [14] and Baş et al. [17], to emphasize novelty.

Comment 2:

Soften causal wording. Change "stress as the major cause" to "stress was frequently reported as a flare trigger.

Response 2:

All causal language has been replaced with association-based phrasing.

Comment 3:

Some in-text references are not linked to citation numbers.

Response 3:

All references now include corresponding numbered citations in the reference list.

Conclusion

Comment 1:

The multidimensional method is valid, but a brief example (e.g., psychological screening + clinical assessment) would improve comprehension.

Response 1:

The final sentence of the Conclusion has been revised to include a concrete example:

“…such as combining BAI-based anxiety evaluation with SDASI assessment in routine visits.”

---

## [Decision Letter · Decision Letter 1]

31 Jul 2025

Thank you for submitting your manuscript to PLOS ONE. After careful consideration, we feel that it has merit but does not fully meet PLOS ONE’s publication criteria as it currently stands. Therefore, we invite you to submit a revised version of the manuscript that addresses the points raised during the review process.

We look forward to receiving your revised manuscript.

Kind regards,

Ahmad Khalid Aalemi, M.D., M.Sc., Ph.D.

Academic Editor

PLOS ONE

Journal Requirements:

Reviewers' comments:

Reviewer's Responses to Questions

**Comments to the Author**

Reviewer #2: All comments have been addressed

Reviewer #3: All comments have been addressed

2. Is the manuscript technically sound, and do the data support the conclusions?

Reviewer #2: Partly

Reviewer #3: Yes

3. Has the statistical analysis been performed appropriately and rigorously?

Reviewer #2: Yes

Reviewer #3: Yes

4. Have the authors made all data underlying the findings in their manuscript fully available?

Reviewer #2: Yes

Reviewer #3: Yes

5. Is the manuscript presented in an intelligible fashion and written in standard English?

Reviewer #2: Yes

Reviewer #3: Yes

Reviewer #2: Conclusions should be rewritten. Conclusions of the abstract in its present form may be written without any study. Conclusions should be based on study results.

Authors mentioned in the Discussion section elevated levels of anxiety in SD patients from two other studies but did not compare their own results with the general population. It can be especially important to support authors' conclusions.

Finally, it is important to reflect the scientific novelty of the study in conclusions.

Reviewer #3: (No Response)

**Do you want your identity to be public for this peer review?** For information about this choice, including consent withdrawal, please see our Privacy Policy

Reviewer #2: No

Reviewer #3: No

---

## [Author Response · Author response to Decision Letter 2]

1 Aug 2025

We thank the Academic Editor and the reviewers for their thoughtful and constructive feedback on our manuscript titled "The psychosocial dimensions of seborrheic dermatitis: A cross-sectional study on anxiety, personality, and quality of life" (Manuscript ID: PONE-D-25-15883R1). We have carefully addressed all comments and revised the manuscript accordingly. In particular, we have:

• Rewritten the Conclusions in both the Abstract and the main text to ensure they are directly grounded in our study results and reflect the scientific novelty of our work;

• Integrated comparative data from previous studies, including control group findings, to contextualize our results with respect to the general population;

• Added a reference to a recent meta-analysis (Chen et al., 2025) to further support the interpretation of our findings;

• Revised the Discussion section to improve clarity, flow, and emphasis on our study's contributions;

• Updated the reference list and re-numbered citations as needed.

Below, we provide a point-by-point response to each reviewer comment, including detailed explanations of all modifications made in the manuscript. We hope that the revised version meets the journal’s standards for publication.

Reviewer comment:

Conclusions should be rewritten. Conclusions of the abstract in its present form may be written without any study. Conclusions should be based on study results. Finally, it is important to reflect the scientific novelty of the study in conclusions.

Author response:

We thank the reviewer for this valuable comment. In accordance with the recommendation, we have completely rewritten the Conclusion sections in both the main text and the Abstract to ensure that they are clearly grounded in the results of our study and reflect the scientific novelty of our work.

Specifically:

1. Generic statements were removed and replaced with data-driven conclusions (e.g., the lack of correlation between SDASI and anxiety, and the moderate correlation between DLQI and anxiety; Rho = 0.465, p < 0.001).

2. We emphasized the dissociation between clinician-rated disease severity and patient-reported burden.

3. We highlighted the novelty of the study, noting that it is among the few to concurrently evaluate clinical, psychological, and personality factors in SD patients.

4. The implications for clinical practice were also clearly stated, including the potential utility of routine psychological screening.

The revised Conclusion (in manuscript) now reads as follows:

This study elucidates the multifactorial nature of seborrheic dermatitis, revealing that the condition is shaped by biological, psychological, and environmental influences. While clinical severity (SDASI) showed no significant associations with anxiety or personality traits, patient-perceived burden (DLQI) was moderately correlated with anxiety severity (Rho = 0.465, p < 0.001), highlighting the dissociation between physician-rated disease and patient experience. These findings underscore the importance of incorporating systematic psychosocial screening into SD management. Routine use of anxiety screening tools (e.g., BAI) alongside conventional clinical assessments may help identify patients in need of psychosocial support, improving patient-centered care. By concurrently evaluating clinical, psychological, and personality dimensions in a single SD cohort, this study contributes novel insights to the dermatological literature and reinforces the value of a multidimensional, biopsychosocial approach in managing chronic inflammatory skin conditions.

The Abstract results and conclusion was also rewritten as follows:

Results

SDASI demonstrated a strong positive correlation with Physician Global Assessment (Rho = 0.815, p < 0.001), confirming its validity as a clinical severity tool. DLQI exhibited moderate positive correlations with both BAI and anxiety severity (Rho = 0.465 and Rho = 0.365, p < 0.001), indicating that anxiety substantially contributes to patient-perceived disease burden. In contrast, SDASI showed no significant associations with anxiety levels or personality traits (all p > 0.05), highlighting a dissociation between physician-rated severity and patient-reported quality of life. Weak correlations were observed between DLQI and female gender (Rho = 0.159, p = 0.011) as well as seasonality (Rho = 0.145, p = 0.018).

Conclusion:

Seborrheic dermatitis imposes a significant psychosocial burden that is more strongly linked to anxiety than to objective clinical severity. Our findings underscore the importance of integrating routine psychological screening—such as BAI assessment—into SD management to identify patients at risk of diminished quality of life. This study contributes novel evidence by concurrently evaluating clinical, psychological, and personality dimensions, reinforcing the need for a multidimensional, patient-centered approach to chronic inflammatory skin disorders.

Reviewer comment:

Authors mentioned in the Discussion section elevated levels of anxiety in SD patients from two other studies but did not compare their own results with the general population. It can be especially important to support authors' conclusions.

Author response:

Thank you for this insightful comment. In response, we have revised the Discussion section to better contextualize our findings in relation to the general population. Specifically, we now reference the study by Cömert et al., which utilized a comparable methodology and reported an anxiety prevalence of 12.6% among healthy controls. In contrast, our cohort exhibited a markedly higher anxiety rate of 30.9%, reinforcing the notion that seborrheic dermatitis is associated with a disproportionate psychological burden.

Additionally, we incorporated the recent systematic review and meta-analysis by Chen et al. (2025), which reported a pooled anxiety prevalence of 19% and depressive symptoms in 21% of SD patients. This further strengthens the argument that the psychological impact of SD exceeds that observed in general or dermatological populations.

To improve overall clarity and cohesion, we also reorganized and edited several sentences in the Discussion. These changes help to better emphasize the novel aspects of our study—namely, the simultaneous evaluation of clinical severity, psychological symptoms, and personality traits.

We note that due to these additions, the reference list has been updated and re-numbered accordingly.

Relevant revision in Discussion section:

Cömert et al., including a control group, found significantly higher anxiety prevalence in SD patients (32.5%) compared to healthy controls (12.6%), but no correlation with disease severity. Our results reflect this pattern, revealing no relationship between anxiety and SDASI, but a moderate correlation with DLQI, reinforcing the distinction between clinical observation and subjective burden. Further supporting these observations, a recent meta-analysis reported clinically significant anxiety in 19% of SD patients—comparable to atopic dermatitis (21%) and acne vulgaris (30%) (18). In our cohort, 30.9% of participants exhibited moderate-to-severe anxiety (BAI ≥16), exceeding the meta-analytic prevalence and closely paralleling the rate reported by Cömert et al. (14). Although our study did not include a control group, these comparisons underscore the substantial psychological burden associated with SD. By simultaneously evaluating clinical severity, psychological distress, and personality traits in a relatively large sample, our study contributes to a more comprehensive understanding of SD and reinforces the need for integrated psychosocial assessment in dermatological care.

---

## [Decision Letter · Decision Letter 2]

11 Aug 2025

The psychosocial dimensions of seborrheic dermatitis: A cross-sectional study on anxiety, personality, and quality of life

PONE-D-25-15883R2

Dear Dr. Gamze Tas Aygar,

We’re pleased to inform you that your manuscript has been judged scientifically suitable for publication and will be formally accepted for publication once it meets all outstanding technical requirements.

Kind regards,

Ahmad Khalid Aalemi, M.D., M.Sc., Ph.D.

Academic Editor

PLOS ONE

Additional Editor Comments (optional):

Reviewers' comments:

Reviewer's Responses to Questions

**Comments to the Author**

Reviewer #2: All comments have been addressed

2. Is the manuscript technically sound, and do the data support the conclusions?

Reviewer #2: Yes

3. Has the statistical analysis been performed appropriately and rigorously?

Reviewer #2: I Don't Know

4. Have the authors made all data underlying the findings in their manuscript fully available?

Reviewer #2: No

5. Is the manuscript presented in an intelligible fashion and written in standard English?

Reviewer #2: Yes

Reviewer #2: (No Response)

**Do you want your identity to be public for this peer review?** For information about this choice, including consent withdrawal, please see our Privacy Policy

Reviewer #2: No

---

## [Editor Report · Acceptance letter]

PONE-D-25-15883R2

PLOS ONE

Dear Dr. taş aygar,

I'm pleased to inform you that your manuscript has been deemed suitable for publication in PLOS ONE. Congratulations! Your manuscript is now being handed over to our production team.

Kind regards,

on behalf of

Dr. Ahmad Khalid Aalemi

Academic Editor

PLOS ONE